# Mechanical Properties of Ca-Saturated Hydrogels with Functionalized Alginate

**DOI:** 10.3390/gels5020023

**Published:** 2019-04-19

**Authors:** Marianne Ø. Dalheim, Line Aa. Omtvedt, Isabel M. Bjørge, Anita Akbarzadeh, João F. Mano, Finn L. Aachmann, Berit L. Strand

**Affiliations:** 1NOBIPOL, Department of Biotechnology and Food Science, NTNU Norwegian University of Science and Technology, N-7491 Trondheim, Norway; marianne.dalheim@ntnu.no (M.Ø.D.); line.a.omtvedt@ntnu.no (L.A.O.); anita.akbarzadeh@ntnu.no (A.A.); finn.l.aachmann@ntnu.no (F.L.A.); 2Department of Chemistry, CICECO, Aveiro Institute of Materials, University of Aveiro, 3810-193 Aveiro, Portugal; isabel.bjorge1@gmail.com (I.M.B.); jmano@ua.pt (J.F.M.)

**Keywords:** Alginate hydrogels, periodate oxidation, reductive amination, mechanical properties, stability, cyclodextrin, peptide

## Abstract

In this work, the mechanical properties and stability of alginate hydrogels containing functionalized alginates (peptide and β-cyclodextrin) were studied. There is an increasing interest in the modification of alginates to add functions such as cell attachment and increased solubility of hydrophobic drugs, for better performance in tissue engineering and drug release, respectively. Functionalization was achieved in this study via periodate oxidation followed by reductive amination, previously shown to give a high and controllable degree of substitution. Young’s modulus and the stress at rupture of the hydrogels were in general lowered when exchanging native alginate with the modified alginate. Still, the gel strength could be adjusted by the fraction of modified alginate in the mixed hydrogels as well as the degree of oxidation. No notable difference in deformation at rupture was observed while syneresis was influenced by the degree of oxidation and possibly by the nature and amount of the grafted molecules. The mixed hydrogels were less stable than hydrogels with only native alginate, and modified alginate was released from the hydrogels. Furthermore, the hydrogels in general rather disintegrated than swelled upon saline treatments.

## 1. Introduction

Hydrogels are 3D-structured water-swollen polymer networks that have been used as scaffold materials in tissue engineering, drug delivery vehicles and in the encapsulation of cells. They are considered suitable biomaterial candidates in these fields as they share many properties with biological tissue [1]. The elastic moduli of biological tissues dependent on the type of tissue [2], and it is, therefore, important to have control over the mechanical properties of the hydrogels used in the various biomaterial applications. Furthermore, the degradability of the hydrogels should also be adjustable. A degradable hydrogel may for example be beneficial if used in tissue engineering, where the goal is tissue regeneration and, hence, replacement of the hydrogel [3]. Alginates form hydrogels by ionic cross-linking of the polymer chains by certain divalent cations such as calcium and barium ions. The gelation can be done at very gentle conditions, e.g., under physiological conditions [4]. Also, alginates show low toxicity and are in general considered as highly biocompatible [5,6], which makes them suitable for in vivo applications.

Alginates are linear polysaccharides that consist of two monomers; β-D-mannuronic acid (M) and its C-5 epimer α-l-guluronic acid (G). It can be isolated from brown algae and certain bacteria. The monomers are 1→4 linked, and can be arranged as stretches of M-units (M-blocks), stretches of G-units (G-blocks), and stretches with alternating M- and G-units (MG-blocks) [4]. The G-blocks are very important for ionic gelation [7,8] and was originally thought to be solely responsible for the selective binding of certain divalent cations (Mg^2+^ << Ca^2+^ < Sr^2+^ < Ba^2+^ [9]). More recent research has shown that also MG-blocks and M-blocks display ion selectivity for Ca^2+^ and Ba^2+^, respectively [10]. Furthermore, the MG-blocks are involved in formation of junction zones upon gelation, either as MG/MG junctions or GG/MG junctions [11]. The composition of the alginate (M/G ratio) and the sequence of M and G residues is important for the mechanical and stability properties of alginate hydrogels. A high G-content is shown to result in mechanically stronger hydrogels compared to alginates with lower G-content [12]. Furthermore, increasing amount and length of MG-blocks have been found to increase both the mechanical strength [11,13] and the stability [12,13] of alginate hydrogels. Recent work has also demonstrated that long G-blocks found in algal alginates (DP > 100) account for their high mechanical strength, in comparison to engineered alginates with similar G content and average length of G-blocks [14]. Alginates varying in composition and sequential structure may be obtained from different sources [15], e.g., different algae or bacteria and different parts of the algae such as stipe and leaf. Furthermore, alginates with defined composition and sequence may be engineered in vitro from mannuronan [16] using a set of isolated bacterial epimerases [12,17].

Alginates may be chemically modified in order to add different functionalities. For example, bioactive peptide sequences, known to bind cellular receptors and induce cellular responses are commonly attached via the carboxyl group of the alginate by carbodiimide chemistry [18,19,20]. We have recently developed an alternative protocol for functionalization, using partial periodate oxidation and reductive amination with 2-methylpyridine borane complex [21] which gives high and controllable degrees of substitution. Previous work has also shown the use of partial periodate oxidation and reductive amination on alginate, but with the use of the toxic reducing agents sodium cyanoborohydride (NaBH_3_CN) or sodium borohydride (NaBH_4_) [22,23,24]. Periodate ions will react with vicinal diols, and in alginates this results in cleavage of the bond between C2 and C3 in the uronic acid residues (ring opening) and formation of an aldehyde group on both C2 and C3 [25]. Various substituents containing primary amines may further be covalently attached to the oxidized residues by reductive amination. Due to the ring opening, the alginate molecule becomes more flexible [26,27,28] and more prone to depolymerization [29] than non-oxidized alginates. The gel-forming properties as well as the mechanical properties of alginate hydrogels is much affected upon partial periodate oxidation (typically 1–8% oxidized residues) [30,31], which is thought to be a combined effect of the above mentioned altered properties as well as disruption of G-blocks and MG-blocks by the oxidized residues. In studies of cell attachment [21] and release of hydrophobic molecules [32] using peptide-grafted and β-cyclodextrin grafted alginates, respectively, prepared via this strategy, the hydrogels were made using a mixture of unmodified and modified alginates. The objective of this study is to further investigate the mechanical properties and stability of hydrogels containing oxidized and functionalized alginates alone and in combination with a strong gel-forming unmodified alginate. Three different substituents varying in size (Figure 1B) as well as in field of application were applied: methyl tyrosine ester as a small size substituent used as a model molecule, the bioactive hexapeptide GRGDSP as a larger substituent relevant for cell attachment and the bulky β-cyclodextrin substituent relevant for drug release. It was hypothesized that the mechanical properties and stability can be adjusted for different applications by varying the fraction of modified and unmodified alginate.

## 2. Results

### 2.1. Preparation and Characterization of the Functionalized Alginates

*L. hyperborea* stipe alginate (hereafter denoted stipe alginate, 65% G, 133 kDa) was used throughout the study as the starting material for modification based on its strong gel-forming properties. The stipe alginate was oxidized using periodate ions according to a final periodate/uronic acid molar ratio (P_0_) = 0.02, 0.04 and 0.08 (reaction scheme in Figure 1A). Periodate oxidation is shown to be stoichiometric for P_0_ up to at least 0.08 [31,33] and a final degree of oxidation corresponding to P_0_ is, therefore, assumed (Table 1). An 8% periodate oxidized stipe alginate (POA, P_0_ = 0.08) was further grafted with methyl tyrosine ester (POA-MeOTyr) and GRGDSP peptide (POA-GRGDSP) obtaining a degree of substitution of 7.0% and 3.9% mole substituent per mole uronic acid residues, respectively (Figure 1B). The general reaction scheme of periodate oxidation followed by reductive amination and the structure of the different substituents is shown in Figure 1A,B, respectively. Furthermore, an 8% oxidized mannuronan was grafted with MeOTyr, giving a degree of substitution of 7.9%. Thereafter, the grafted mannuronan was epimerized (epim POA-MeOTyr) with AlgE64, giving a G content of 49%. The obtained G-content is similar to a previous study were AlgE64 was used to epimerize partially oxidized (P_0_ = 0.08) and reduced mannuronan [34].

β-CyD was grafted to POA (P_0_ = 0.08) in two steps. First, a linker (4-pentyn-amine) was coupled to the oxidized alginate by reductive amination (Figure 1A,B) giving a degree of substitution of 1.6%, upon which pentyn amine coupled POA was fully grafted with β-CyD via the copper catalyzed click-reaction (Figure 1C) giving a final degree of substitution of 1.6%.

The degree of substitution was throughout this work analyzed by ^1^H NMR spectroscopy (Appendix A) and calculated as mole substituent per mole uronic acid residues as previously described [21,32]. The calculations are based on earlier annotations of the proton NMR spectra of alginates [35,36], periodate oxidized alginates [31], MeOTyr grafted alginates [21], GRGDSP grafted alginates [20] and β-CyD grafted alginates [32]. The degree of substitution obtained here is in well accordance with what we have previously reported using the same methods for grafting of stipe alginate [21,32].

The molar mass of the materials was analyzed using SEC-MALS (Appendix A) and the obtained average molecular weights (*M*_w_) are summarized in Table 1. The opening of the sugar ring makes oxidized alginates more prone to degradation via β-elimination and as previously observed, the oxidation was accompanied by some degradation [21,29]. A slight increase in M_w_ was observed upon grafting of MeOTyr and the peptide GRGDSP that, taking into account the added weight of the substituents, indicate minimal degradation during grafting. Grafting of β-CyD was, on the other hand accompanied with a reduction of M_w_ from 97 to 65 kDa, despite the high molecular weight substituent. Depolymerization was predominantly happening during grafting of the pentyn-amine linker [32], which can be attributed to the low degree of substitution. Grafting has been shown to reverse the degradability of the oxidized alginate back to that of unmodified alginate [37] and degradation is therefore, more pronounced during grafting of pentyn-amine compared to the more efficient reactions with MeOTyr and GRGDSP.

### 2.2. Ca-Saturated Alginate Hydrogels of Functionalized Alginates

Alginate hydrogels were prepared from the different modified alginates (Figure 2A–F). The hydrogels were saturated with calcium ions by dialysis, implying that maximum crosslinking of the alginate within the hydrogel was obtained. Oxidized alginates having 2% and 4% oxidized residues formed nice gels that maintained their shape upon handling. For the 8% oxidized material, on the other hand, the gel cylinders were largely affected by the oxidation and only very weak and largely deformed hydrogels could be formed. Ca-hydrogels of the epimerized POA-MeOTyr formed highly syneretic gels that showed a higher degree of shape stability after gel formation (Figure 2E) compared to the other hydrogels made of only 8% periodate oxidized alginate.

Young’s modulus, syneresis, stress and deformation at rupture were measured on the oxidized and grafted materials by uniaxial compression measurements (Table 2, Appendix A). In general, the syneresis was high, but less for oxidized and grafted materials than for the only oxidized material. However, Young’s modulus was low for all the hydrogels. This was also the case for the stress and deformation at rupture, i.e., for the highest deformation before the gels failed and at the force at the same deformation (see example in Appendix A). However, both Youngs modulus and stress at rupture increased when MeOTyr was grafted to the alginate, in contrast to β-CyD that resulted in decreased gel strength. The epimerized alginate showed a slightly higher Young’s modulus than the corresponding hydrogels of oxidized and grafted alginates of stipe alginate. However, the stress at rupture for the epimerized alginate gels were 5–6 times higher than for the corresponding gels of POA-MeOTyr made from stipe alginate.

### 2.3. Mechanical Properties of Mixed Ca-Gels of Oxidized Alginate and Stipe Alginate

To investigate the effect of varying the fraction of modified alginate in the hydrogel as well as the materials’ degree of oxidation, Ca-saturated hydrogels were prepared from mixtures of stipe alginate (F_G_ = 0.65) and oxidized alginates (POA) in different ratios. Also, Ca-limited gels were investigated for gelling kinetics and rheology. The total alginate concentration in both cases was 1.0% *w*/*v*, unless otherwise specified.

In general, the Ca-saturated hydrogels prepared from mixtures with stipe alginate resulted in nicely formed gels that maintained their shape upon handling (see Figure 2D). Oxidation of the alginate led in general to a reduction in Young’s modulus (*E*) and stress at rupture, whereas syneresis and deformation at rupture were less affected by the oxidation (Figure 3). For hydrogels where the weight fraction of POA (W_POA_) was 0.25, Young’s modulus was not significantly different from the pure stipe alginate gel (see Appendix A), varying around 30 kPa for all three degrees of oxidation. Mixing in oxidized alginate where 2% of the residues had been oxidized (P_0_ = 0.02) did not have a large effect on E, while hydrogels prepared from this material alone (W_POA_ = 1.00) showed a substantial decrease in the Young’s modulus of 9 kPa. The hydrogels containing POA prepared with P_0_ = 0.04 showed a more gradually decrease in E, while for the hydrogels composed of different amounts of POA P_0_ = 0.08 a substantial decrease in E was observed for gels having 50 to 75% POA (dry mass).

Hydrogels with various fractions of 4% and 8% oxidized alginate were in general less syneretic than the stipe alginate reference. The syneresis decreased with increasing degree of oxidation, and to a certain extent with increasing amount of oxidized alginate (W_POA_, Figure 3B). In comparison, decreasing the total alginate concentration resulted in a higher degree of syneresis, as shown for 0.5% *w*/*v* stipe alginate hydrogels in Figure 3B. For the mixed gels with POA P_0_ = 0.02, the degree of syneresis was not largely affected by the relative amount of POA.

The deformation at rupture (%) was only slightly reduced in the mixed hydrogels, relative to the stipe alginate reference hydrogel (Figure 3C). The force needed to rupture the gels (stress at rupture) was, on the other hand, highly dependent on the relative amount of POA as well as the degree of oxidation (Figure 3D). Mixing in POA resulted in a lower stress at rupture that decreased with increasing degree of oxidation and with the relative amounts of POA (Figure 3D).

Some selected mixed gels with stipe alginate and 8% oxidized alginate (Figure 4) were further studied in a rheometer where the gelling kinetics were followed over time by recording shear modulus upon release of Ca^2+^ from CaCO_3_ following hydrolysis of glucono-*δ*-lactone (GDL) [38]. Ca-limited gels were chosen due to limited syneresis and, hence, good contact between the gel and the probe, as well as the general relevance of Ca-limited gels as softer gel materials than Ca-saturated gels. As expected, the shear modulus (G’) decreased upon exchange of stipe alginate with oxidized alginate from 1205 to 535 Pa and 227 Pa for stipe alginate in 50% and 75% mixture with oxidized alginate, respectively. The gelling kinetics were slower for the mixed gels than for the stipe alginate alone, where 8–12 h were needed to reach plateau levels for the mixed gels versus 4–6 h for the stipe alginate.

### 2.4. Mechanical Properties of Mixed Ca-Saturated Hydrogels with Functionalized Alginate

Ca-saturated hydrogels of 8% oxidized alginate as well as 8% oxidized alginate grafted with MeOTyr, β-CyD and GRGDSP were further characterized to investigate the effect of the grafted groups on the mechanical properties of the hydrogels (Figure 5). The weight fraction of functionalized POA (W_POA*_) was varied between 0.00 and 0.75. All materials were oxidized to 8%, but the final degree of substitution varied for the different grafted molecules (see Table 1).

Coupling of MeOTyr, β-CyD or GRGDSP to the oxidized residues resulted in a lower Young’s modulus compared to hydrogels with similar amounts of POA for W_POA*_ = 0.25 and 0.50. The different substituents showed similar effect on Young’s modulus, independent of their degree of substitution. At W_POA*_ = 0.75, no difference was observed compared to hydrogels with 75% w/w POA. Neither deformation nor stress at rupture was largely affected by grafted alginates relative to POA. With regards to the deformation at rupture, only the hydrogels grafted with MeOTyr showed a significant difference compared to the POA hydrogels at W_POA*_ = 0.50 (see Appendix A), and for the stress at rupture the hydrogels grafted with MeOTyr showed a significant difference compared to the POA hydrogels at W_POA*_ = 0.25 (Appendix A). POA-MeOTyr and POA-GRGDSP showed a higher syneresis compared to POA. This was not observed for POA-β-CyD that had lower or similar syneresis to the oxidized alginate. The epimerized alginate grafted with MeOTyr behaved similarly to the stipe alginate grafted with MeOTyr in mixture with non-modified stipe alginate (W_POA*_ = 0.50), and no significant difference was found between the hydrogels except with regards to the deformation at rupture data assuming that *p*-values less than 0.05 are significant (see Appendix A for *p*-values).

### 2.5. Stability of Mixed Hydrogels and Leakage of Material upon Saline Exposure

The stability of stipe alginate/POA (P_0_ = 0.08) and stipe alginate/POA-MeOTyr mixed hydrogels was studied with regards to Young’s modulus, swelling and leakage material through a series of consecutive saline treatments (0.15 M NaCl, 24 h each) (Figure 6). Young’s modulus (E) was gradually reduced for every saline treatment for all hydrogels studied. For all the different mixed hydrogels, E decreased gradually with increasing number of saline treatments, approaching about 2 kPa before the gels lost their cylindrical form and hence, it was not possible to perform compression measurements (at 2–4 treatments of saline). Hydrogels with 75% POA or MeOTyr grafted POA was difficult to handle after the second treatment. At all points, the highest Young’s modulus value was obtained for the stipe alginate gels, which, furthermore, showed only slightly reduced structural integrity upon the fourth saline treatment.

The swelling, which we characterized as the relative weight gain, is shown in Figure 6A. In general, minor swelling occurred upon the first saline treatment, after which it increased slightly before the hydrogel dissolved. The exception was gels containing 75% POA, for which the weight decreased upon the saline treatments due to disintegration of the gels.

As expected from the stability data, the mixed hydrogels had a higher amount of leaked material upon saline treatments than the stipe alginate gel (Figure 6C). A small amount of material leaked from the stipe alginate hydrogels upon the first four saline treatments (1–7%). In the fourth treatment, the gels started to lose their structure. In comparison, the mixed gels leaked more material and could withstand fewer saline treatments. The leakage was highest for gels with the highest content of modified alginate, e.g., 75% *w*/*w* POA or POA-MeOTyr with 44 and 52 wt% total leaked material in the first and second treatment for gels with POA or POA-MeOTyr, respectively.

The leaked material was further characterized using ^1^H NMR spectroscopy and SEC-MALS in regards of composition and molecular weight, respectively (Figure 7). Initially, low molecular weight molecules leaked out of the hydrogels however, M_w_ increased gradually in the subsequent saline treatments approaching the M_w_ of the stipe alginate starting material (133 kDa). The leaked alginate from hydrogels with 75% *w*/*w* POA or POA-MeOTyr had a higher M_w_ than for the other gels (87 kDa for W_POA_ = 0.75 compared to 32 kDa for W_POA_ = 0.50 in the first saline treatment).

The alginate that leaked from the mixed hydrogels during the first saline treatment had a low guluronic acid content (F_G_~0.4–0.5) compared to that of the starting material (F_G_ = 0.65 for the stipe alginate). The G-content then increased, reaching that of the starting material in the second or third treatment. Similarly, the average G-block length of the leaked alginate tended to increase upon the consecutive saline treatments. The amount of leaked material from the stipe alginate hydrogels during the first saline treatment was too small for further analysis. In the subsequent leakage materials, both the G-content and average G-block length corresponded to that of the stipe alginate.

The MeOTyr coupled alginate used in this study had a degree of substitution of 7.0%, hence, a fraction of uronic acid residues coupled to a MeOTyr molecule (F_MeOTyr_) of 0.07. NMR analysis of the leaked material from hydrogels with W_POA-MeOTyr_ = 0.50 and 0.75 indicated that the leaked alginate in the first saline treatment had a higher F_MeOTyr_ than the starting material (0.11 and 0.10 for W_POA-MeOTyr_ = 0.50 and 0.75, respectively). In the consecutive treatments, F_MeoTyr_ decreased, reaching 0.05 in the third saline treatment of mixed hydrogels with W_POA-MeOTyr_ = 0.50.

## 3. Discussion

The use of functionalized alginates (e.g., peptide grafted alginates) in alginate-based biomaterials is increasing and it is generally accepted that various tissue engineering- and drug release strategies will have different requirements regarding the stability and mechanical properties of the applied hydrogels. The differentiation of stem cells can for example be affected by mechanical properties [39]. The data presented here provide an increased understanding of how functionalization via periodate oxidation and subsequent grafting with peptide or cyclodextrins affect the mechanical properties and stability of alginate hydrogels. Furthermore, the data show, in principal, how mechanical properties of chemically modified alginate in mixtures with a strong gel forming alginate can be adjusted by varying the extent of modification and relative amount of functionalized alginates. Alginate with 8% oxidation was used for the grafting studies to better examine the differences between hydrogels made of both modified alginate and non-modified alginate, while also giving the benefit of a higher degree of functionality compared with a starting point of 2 or 4% oxidation.

### 3.1. Mechanical Properties

Rheological and mechanical studies on hydrogels composed of only periodate oxidized alginate has been performed by several groups, including ours [30,31,40]. The alginate characteristics (composition, G-block length, degree of oxidation and molecular weight) and Ca^2+^ concentration varied in the different studies, yet the general conclusion is that the ability to form hydrogels as well as the mechanical properties is highly affected upon partial oxidation (1–10%), and that increasing the degree of oxidation results in lower elastic moduli (G’) [30,31]. Periodate ions are known to react faster with G-residues than M-residues (approximately 50% faster) [41], leading to a relatively larger influence on the G- and MG-blocks than on M-blocks. Previous studies have shown that the G-block length is reduced when using periodate oxidation on alginates [31]. Hence, oxidized alginates are expected to contain shorter G-blocks and consequently form shorter junction zones with the gelling ions. It is known that for alginate hydrogels, the shortest G-block length to form stable junctions with Ca^2+^ is 8 [42,43], whereas Sr^2+^ only requires a G-block length of 3 [42]. Hence, periodate oxidation may reduce the length of consecutive G units to below the critical values for crosslinking with calcium ions. A more in-depth characterization of the local hydrogel structure, in regards of size and multiplicity of the junction zones is possible by scattering techniques such as small angle X-ray scattering [44,45]. This is however beyond the scope of our work.

The mechanical properties were here characterized by Young’s modulus (E), which is dependent on the number, stability and length of crosslinks (junction zones) as well as the length and flexibility of the elastic segments between junction zones [46]. Some selected materials were also characterized by rheology for Ca-limited gels, where shear modulus (G´) is proportional to E. Ca-saturated hydrogels where the weight fraction of POA (W_POA_) was 0.25 or 0.50, showed similar or slightly reduced Young’s modulus’, respectively, compared to the reference stipe alginate gel for all degrees of oxidation. In comparison, reducing the total alginate concentration from 1.0% to 0.5% *w*/*v* in stipe alginate gels resulted in a large decrease in Young’s modulus from 31 kPa to 6 kPa. It follows that, despite the partial disruption of G- and MG-blocks, the oxidized alginate makes a substantial contribution to the junction zones. A large decrease in Young’s modulus was observed using W_POA_ = 0.75 of the 8% oxidized alginate. Mixing ratios from 0.50 to 0.75 could then be exploited to give a range of mechanical strengths. For the calcium-limited gels (unsaturated gels), mixing ratios of 0.50 and 0.75 resulted in 56% and 81% reduction in shear modulus (G´, Figure 4), respectively, that may point to a further reduction in gel strength in mixed gels when calcium is limited. The 8% oxidized alginate did not form gels under Ca-limited conditions and formed very weak gels upon Ca-saturation without the support of the non-modified alginate, further demonstrating the reduced ability to form strong junction zones upon introduction of the oxidized residues. Interestingly, coupling MeOTyr to the oxidized alginate resulted in slightly higher Youngs modulus and stress at rupture compared to the oxidized material only. This may be due to hydrophobic interactions between MeOTyr contributing to the mechanical properties of the gel.

Grafting resulted in a reduction in Young’s modulus for W_POA_ 0.25 and 0.50 relative to the oxidized alginate. In the mixed gels, the potential contribution from hydrophobic interactions is probably reduced due to steric hindrance with the presence of non-modified alginate. Addition of free β-CyD did not have the same effect as grafted β-CyD (see Figure 5), suggesting that covalent coupling induce steric restrictions that negatively affect the number of junction zones formed in the mixed gels. No difference was observed for the different substituents (see Appendix A for statistical values), despite that the degree of substitution ranged from 1.6% for POA-β-CyD to 7.0% for POA-MeOTyr (both with P_0_ = 0.08). The lower molecular weight of POA-β-CyD may, however, limit its contribution to interchain junctions [47], resulting in a reduction of gel strength despite the low degree of substitution.

The stress at rupture was substantially lowered by both increasing the fraction of POA and increasing the degree of oxidation. Rupture strength is directly linked to the energy required to fracture the network junctions, and shorter junction zones in periodate oxidized alginates concomitantly result in a lower stress at rupture. Grafting of substituents to POA did not further decrease the force needed to induce rupture, suggesting that the substituents do not affect the strength of the junction zones.

The deformation at rupture, measured as the length of compression relative to the initial height of the hydrogels (%), is a measure for how much deformation the hydrogel can withstand before rupture. All the mixed hydrogels were largely compressed to the same level and was thus minimally affected by the relative amount of POA, the degree of oxidation of the incorporated oxidized alginate nor the introduction of substituents. However, as lower force was needed for them to rupture, alginates functionalized through periodate oxidation and reductive amination form more fragile hydrogels.

Using a chemoenzymatic strategy, by introducing the chemical modifications on mannuronan and secondly introducing the G-blocks by epimerization, has previously been shown to be efficient in maintaining the mechanical properties of the Ca–alginate hydrogels [12]. The same strategy has been used for the grafting of bioactive peptides to alginate using carbodiimide chemistry [20]. Mannuronan C-5 epimerases are shown to work on both oxidized alginates as well as alginate in the gel state [34]. In this work, the grafted alginate produced with a chemoenzymatic strategy (EpimPOA-MeOTyr) was hypothesized to give better mechanical properties compared to the non-epimerized POA-MeOTyr due to the grafting only taking place on the M-residues. Here, the EpimPOA-MeOTyr showed better gelling properties relative to the corresponding grafted stipe alginate (POA-MeOTyr), resulting in more defined gels and with stress at rupture close that to unmodified stipe alginate (3 versus 4 kg, respectively). However, Young’s modulus was low and comparable to the corresponding POA-MeOTyr made of stipe alginate (Table 2). For mixed hydrogels, the unmodified stipe alginate dominated the mechanical properties and no significant differences were seen between the grafted epimerized alginate and the corresponding POA-MeOTyr made of stipe alginate.

### 3.2. Syneresis

Ionically crosslinked alginate gels are in general smaller than the starting volume of the liquid that was used to cast the gel. The negative change in volume upon gel formation is referred to as syneresis. The main driving force of syneresis is thought to be the growth of junction zones [48] and release of water. In alginates, both the length of the G-blocks and the presence of alternating sequences (MG-blocks) will affect the syneresis. Long G-blocks give little syneresis as the network is not easily reorganized [47]. Alternating sequences are, on the other hand, associated with high syneresis [12,17]. This has been attributed to the formation of MG/MG junctions leading to a partial network collapse upon Ca^2+^ saturation [13]. Hydrogels containing POA oxidized to 4 and 8% expressed less syneresis than the stipe alginate reference gel and the syneresis decreased slightly with increasing amount of the oxidized alginate. Opening of the sugar ring upon periodate oxidation increases the local flexibility, which could promote reorganization and, hence, syneresis, but reduces the effective junction zones, possibly both in number and length. Hence, it seems like oxidized alginates have a reduced ability to contribute to growth of effective junction zones that surpass the increased flexibility. An oxidation degree of 2% showed similar syneresis to the unmodified stipe alginate hydrogel, indicating that this low oxidation degree did not alter the properties of the modified alginate compared to the parent alginate markedly. A lower total concentration of alginate in the pre-gel solution gave as expected higher syneresis [49] (shown for 1.0% and 0.5% *w*/*v* stipe alginate hydrogels in 3B). Mixing-in oxidized alginate can therefore, not be seen as an effective decrease in concentration.

Grafting of molecules to the oxidized residues could possibly add steric hindrance during network reorganization and consequently reduce the syneresis. For hydrogels with 50% *w*/*w* POA or functionalized POA, the opposite was observed and syneresis increased in the order POA-β-CyD ≈ POA < POA-GRGDSP < POA-MeOTyr. This order corresponds well with the degree of substitution for the different materials (POA-β-CyD < POA-GRGDSP < POA-MeOTyr), indicating that higher level of grafting leads to a more compact hydrogel. Syneresis is known to increase with an increasing concentration of the crosslinking ion and molecular weight of the alginate [47,48]. The substituents could possibly affect the diffusion of calcium ions or exclusion of water and hence, syneresis, or there may be interactions between the substituents leading to a more compact gel network. POA-β-CyD was more degraded during preparation than the other alginates (*M*_w_ = 65 kDa, compared to 114 kDa for POA-MeOTyr), which together with the low DS (1.6%, Table 1) can account for the observed syneresis being similar to or slightly lower than POA.

### 3.3. Stability and Leakage

Alginate gel stability in saline solution is relevant for the use and handling of alginate gels and for the use at physiological conditions. The optimal stability of an alginate hydrogel depends on the application. For slow release of a drug from an alginate hydrogel, a robust long-lasting gel could be beneficial. On the other hand, for some tissue engineering applications, a less stable hydrogel which is more readily replaced with tissue would be better suited [3]. Ca–alginate hydrogels are vulnerable to destabilization upon exposure to saline due to the exchange of calcium with sodium ions that result in destabilization of the gelling zones and leakage of alginate. Swelling of Ca–alginate gels is previously described by the balance of exchange of Ca^2+^ with Na^+^ and influx of water and the gel network ability to withstand the osmotic pressure [17,50]. The stability of the mixed hydrogels was investigated as the change in Young’s modulus and weight (swelling) and leakage of alginate upon consecutive saline treatments. A gradual decrease in Young’s modulus was observed for all the different gel types, reflecting the gradual exchange of Ca^2+^ with non-crosslinking Na^+^ [17] resulting in a destabilization of the junction zones. To create more stable and long-lasting hydrogels, the gelling ions strontium or barium could be introduced as these ions bind more strongly to the G-blocks in the alginate and are depending on shorter G-blocks to form stable crosslinks [10]. In contrast, pre-treatment of the gels in saline bath could be a means for achieving faster degradation.

The mixed hydrogels were less stable than stipe alginate gels as they were able to withstand fewer saline treatments. As the total concentration of alginate was the same in all gels and, hence, also the charge density, the exchange of Ca^2+^ with Na^+^ and flux of water is expected to be approximately the same in the hydrogels. However, the reduction in G-blocks and, hence, crosslinking leads to a destabilization of the network. The mixed hydrogels and the stipe alginate gels were disintegrating rather than the typical swelling, observed as increase in volume and weight, previously seen for alginate gel beads upon consecutive saline treatments [10,17]. In addition, large amounts of alginate were released (leaked) from the mixed gels compared to the stipe alginate gels. A small amount of swelling and leakage of alginate from Ca-hydrogels from *L. hyperborea* alginate with similar composition and sequence parameters as the stipe alginate used here has been reported [13,42]. Also, the leakage material from previous studies was shown to contain shorter chains of high M containing alginate [42]. Here, an accumulation of modified alginate in the leakage material was seen, which is connected to the destruction of the crosslinking zones in the grafted alginate.

## 4. Conclusions

The mixed hydrogels with POA and *L. hyperborea* stipe alginate gave lower Young’s modulus and stress at rupture than control gels made of unmodified *L. hyperborea* stipe alginate only, but the mechanical properties could be controlled by the fraction of modified alginate in the hydrogels as well as the degree of oxidation. Grafted epimerized alginate showed better gelling properties (higher Young’s modulus and deformation at rupture) than the corresponding grafted stipe alginate. For hydrogels with a mixture of modified and unmodified stipe alginate, the mechanical properties were largely influenced by the unmodified stipe alginate. Decreased syneresis was observed upon increasing the degree of oxidation, but was only slightly affected by the relative amount of the modified alginates. Also, the data suggests that syneresis was influenced by the nature and amount of the grafted molecules. The mixed hydrogels were less stable than gels with only unmodified alginate, and the grafted chains were leaking out first. Furthermore, the mixed hydrogels were found to disintegrate rather than swell upon saline treatments. It is evident that functionalization via periodate oxidation and subsequent grafting (via reductive amination and Cu-catalyzed click-chemistry) has considerable effect on alginates gelling properties. This must be taken into account, and can be utilized upon designing alginate hydrogels as biomaterials.

## 5. Materials and Methods

### 5.1. Materials

Alginate from *Laminaria hyperborea* stipe was obtained from FMC Health and Nutrition, Drammen, Norway. Its composition obtained from NMR spectroscopy and molar mass (*M*_w_) obtained from SEC-MALS are given in Table 3. The hexapeptide GRGDSP was purchased from CASLO ApS, Denmark and 6-*O*-monodeoxy-6-monoazido-β-cyclodextrin (N_3_-β-CyD) was synthesized in-house [51] by Thorbjørn Nielsen at Aalborg University. All other chemicals were purchased from commercial sources and were of analytical grade. Mannuronan was obtained from FMC Health and Nutrition ([η] = 1548 mL/g), F_G_ =0.00.

### 5.2. Periodate Oxidation and Preparation of Functionalized Alginates

Stipe alginate and mannuronan were oxidized upon addition of sodium periodate as previously described [31]. l-Tyrosine methyl ester (MeOTyr) was grafted to both the oxidized stipe alginate and mannuronan as previously described [21]. In addition, the peptide GRGDSP was grafted to stipe alginate in the same manner. In short, the alginates were oxidized using a concentration of periodate ions that corresponded to a P_0_ of 0.08. The periodate oxidized alginates were further reacted with MeOTyr or GRGDSP using 2-methylpyridine borane complex (picoline borane) as the reducing agent.

β-CyD was grafted to periodate oxidized alginate (P_0_ = 0.08) in two steps, as previously described [32]. A linker, 4-pentyn-1-amine, was first grafted to the periodate oxidized alginate by reductive amination. Thereafter, β-CyD was linked to 4-pentyn-1-amine coupled alginate via the Cu(I)-catalyzed azide alkyne cycloaddition click-reaction [51].

The l-Tyrosine methyl ester grafted mannuronan was epimerized with AlgE64 [34] to obtain an alginate were the G-units were introduced into the chain after the grafting procedure. Briefly, MeOTyr grafted mannuronan was dissolved in water, and the enzyme and epimerization buffer were then added. The ratio between the enzyme and grafted alginate used was 1 mg enzyme per 25 mg grafted mannuronan. The final concentrations in the reaction mixture were 2.4 mg/mL grafted mannuronan, 50 mM MOPS, 75 mM NaCl and 2 mM CaCl_2_. The reaction mixture was incubated at 37 °C, and after 48 h the reaction was stopped by adding 10 mL EDTA (100 mM, pH = 7). The product was dialyzed until conductivity measured below 1 µS.

### 5.3. Preparation of Calcium–Alginate Hydrogels

Internally set calcium alginate gel cylinders (Ca-limited or unsaturated gels) were prepared by adding CaCO_3_ and the slowly hydrolyzing d-glucono-*δ*-lactone (GDL), as previously described [38]. Briefly, the release of proton from GDL and the subsequent lowering of pH destabilize the CaCO_3_ and cause release of calcium ions that will bind to the alginate. Alginates were dried overnight in a desiccator with phosphorus pentoxide (Sicapent, Sigma-Aldrich, Oslo, Norway) and dissolved in MQ water. A dispersion of CaCO_3_ (4 μm, ESKAL 500, KSL Staubtechnik GMBH, Lauingen, Germany) was added to the alginate solutions and the solutions were thereafter degassed, using vacuum suction. GDL was then added to the mixtures to start the gel formation. The mixtures were stirred carefully for a few seconds before they were poured into the wells of a 24-well plate (16/18 mm, Costar, Cambridge, MA, USA) and left at room temperature for 20 h. The final concentrations were 1.0% (or 0.5%) *w*/*v* alginate, 15 mM CaCO_3_ and 30 mM GDL. The amount of calcium added is calculated to theoretically crosslink all G present, however, it is shown that the gel strength and syneresis of calcium gels further increase with calcium concentration [38]. Hence, Ca-saturated hydrogels were made by immersing the gel cylinders in 50 mM CaCl_2_ and 0.2 M NaCl (800 mL for 8 cylinders) for 24 h at 4 °C to ensure excess of calcium in the gels.

### 5.4. Syneresis and Mechanical Properties

The syneresis (the volume of gels relative to the initial volume of solute) was determined as the weight reduction of the calcium-alginate hydrogels with respect to the initial weight and calculated according to S = (1 − *w*/*w*_0_) × 100, where w_0_ and w is the initial and final weight respectively. The initial weight was calculated based on the volume of the wells, assuming no significant change in density from water for 0.5–1.0% alginate.

The mechanical properties of the alginate gel cylinders were characterized by uniaxial compression using a Stable MicroSystems, TA XT plus Texture Analyser with a P/35 probe at room temperature and a compression rate of 0.1 mm/s. The weight of the loading cell was five kg. Young’s modulus (E) was calculated from the initial slope (typically 0.1–0.3 mm) of the force/deformation curves (F/A) = E × ∆l/l (l = length of gel before measurement, ∆l = change in length) and corrected for syneresis: E_corrected_ = E_measured_ × (C_initial_/C_final_)^2^ [52]. All data were collected and processed with the “Texture Expert Exponent 32” software. Stress at rupture and deformation at rupture (or ultimate compression strength) was determined in experiments where the hydrogels were compressed to the point of rupture, and the reported data are the maximum value on the force-deformation curve (See Appendix A).

### 5.5. Rheological Characterisation of Unsaturated Gels

1.0% (*w*/*v*) Ca-limited (unsaturated) alginate gels were prepared as described above (4.3) by internal gelation with final concentration of 15 mM CaCO_3_ and 30 mM GDL. Immediately after addition of GDL and rapid and careful mixture to avoid bobble formation, 0.65 mL sample was loaded to a rheometer and recordings started 10 min after adding GDL to the alginate solution. The development of shear modulus (G´, Pa) was followed over time in room temperature by Kinexus Rheometer (Malvern instruments, Uppsala, Sweden) using a 2 cm flat probe and a flat plate geometry with 1 mm gap and recorded at a frequency of 1 Hz.

### 5.6. Stability

The stability of the hydrogels was analyzed through a series of consecutive saline treatments of Ca-saturated gel cylinders. For each gel type, five gels were immersed in 100 mL 0.15 M NaCl (20 mL per gel) and left for 24 h at 4 °C with gentle stirring (first saline treatment). Swelling is defined as volume increase, however, as alginate gels increase in volume by absorbing the surrounding solution that correspond to subsequent increase in weight, swelling of the gels were characterized as *w*/*w*_0_, where *w*_0_ is the initial weight before the first saline treatment and w is the weight after the respective saline treatments. Compression measurements (1 mm compression) were then performed before the hydrogels were immersed in a new 0.15 M NaCl solution (second saline treatment). This was repeated until the gels had lost their cylindrical form and it was not possible to perform mechanical measurements on them (2–4 treatments).

### 5.7. Analysis of the Leaked Alginate

After removing the hydrogels, the separate NaCl treatment solutions was added 2 mL of a 150 mM EDTA solution. The saline treatments solutions were further dialyzed, using a MWCO of 3.5 kDa against two shifts of 0.05 M NaCl and then MQ water until the conductivity was below 4 μS/cm. The samples were freeze dried and further prepared for NMR and SEC-MALS analysis as described below.

### 5.8. NMR

1D ^1^H NMR spectroscopy was used to analyze the degree of substitution as well as the uronic acid composition of the alginates. The samples were subjected to mild acid hydrolysis as previously described [53] to reduce the viscosity prior to the NMR analysis. 6–10 mg of the samples were then dissolved in 600 µL D_2_O (99.9%) and added 5 µL 3-(Trimethylsilyl)propionic 2,2,3,3-*d*_4_ acid (TSP, Sigma-Aldrich, Oslo, Norway) as an internal standard and 15 µL Triethylenetetraamine-hexaacetic acid (TTHA, Sigma Aldrich) as a chelating agent for residual divalent ions. The latter was not added to samples already added EDTA (analysis of leaked alginate). 1D ^1^H spectra were recorded at 90 °C on a Bruker Ascend 400 MHz Avance III HD spectrometer, equipped with a 5-mm SmartProbe z-gradient probe and SampleCase (Bruker BioSpin AG, Fällanden, Switzerland). The spectra were recorded using TopSpin 3.2 software (Bruker BioSpin, Fällanden, Switzerland) and processed and analyzed with TopSpin 3.5 software (Bruker BioSpin).

### 5.9. SEC-MALS

Size exclusion chromatography with multi angle light scattering (SEC-MALS) was used to measure the molecular weight, as previously described [28]. The system consisted of a mobile phase reservoir, an on-line degasser, an HPLC isocratic pump (LC-10ADVP, Shimadzu, Kyoto, Japan), an autoinjector (SCL-10AV, Shimadzu), a pre-column and two serially connected columns: TSK 6000 PWXL and TSK 5000 PWXL (Toso Haas, Stuttgard, Germany). The columns were followed by two serially connected detectors; a Dawn DSP laser light scattering photometer (Wyatt Technology, Santa Barbara, CA, USA) (λ_0_ = 0.664 nm) and an Optilab DSP differential refractometer (Wyatt Technology, Santa Barbara, CA, USA). 0.15 M NaNO_3_ with 0.01 M NaEDTA, pH 6 was used as the mobile phase. 20% acetonitrile was added to the mobile phase for the alginate grafted with β-CyD. All samples were dissolved in the mobile phase (0.25–7.5 mg/mL) and filtered (pore size 0.8 µm) prior to injection. The injection volume was adjusted to obtain an optimal light scattering signal and to avoid overloading. Astra software v. 6.1 (Wyatt Technology, Santa Barbara, CA, USA) was used to collect and process the obtained data, using a refractive index increment (dn/dcµ) of 0.150 mL/g for alginate samples [28].

### 5.10. Statistical Analysis

The Tukey honest significant differences test was done using R to examine significant differences between the hydrogels with regards to the Young’s modulus, syneresis, deformation at rupture and stress at rupture measurements. For the comparison of the POA-MeOTyr gels with regards to epimerized and non-epimerized alginate the Welch *t*-test was used to compare the two sets of gels. The statistical values can be found in Appendix A. Data is shown as mean ± standard deviation. Samples with *p*-values less than 0.05 were considered statistically significant.

## Figures and Tables

**Figure 1 gels-05-00023-f001:**
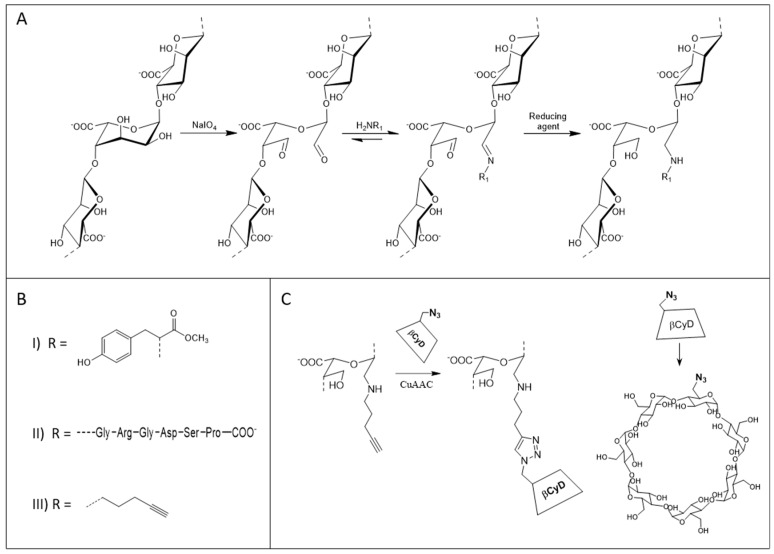
(**A**): General reaction scheme for periodate oxidation (step 1) followed by reductive amination (step 2 and 3) of alginate (here represented by an MGM-fragment). (**B**): Structure of the different substituents grafted onto alginate by reductive amination. I: l-methyl tyrosine ester (MeOTyr), II: the hexapeptide GRGDSP, III: 4-pentyn-amine, used as a linker for further grafting of β-cyclodextrin (β-CyD). (**C**): Grafting of β-CyD to pentyn-amine linked alginate via the copper catalyzed azide alkyne cycloaddition click-reaction. The scaling of alginate and β-CyD is not similar.

**Figure 2 gels-05-00023-f002:**
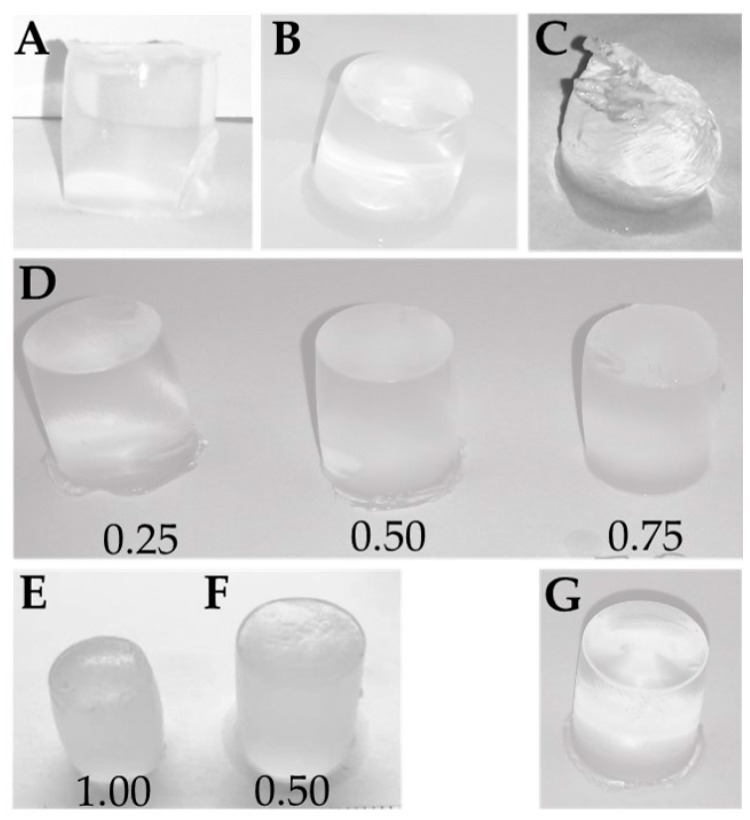
Ca-saturated alginate hydrogels (1.0% *w*/*v*). (**A**): POA (P_0_ = 0.02), (**B**): POA (P_0_ = 0.04), (**C**): POA (P_0_ = 0.08), (**D**): Mixed hydrogels of POA (P_0_ = 0.08) and stipe alginate in fractions of W_POA_ = 0.25, 0.50 and 0.75. (**E**): Epim POA-MeOTyr solely, (**F**): Mixed hydrogels of Epim POA-MeOTyr and stipe alginate (W_POA*_ = 0.50). (**G**): Hydrogels of solely stipe alginate. Abbreviations: POA: Partially periodate oxidized alginate. P_0_: Fraction of oxidized residues (degree of oxidation). Epim POA-MeOTyr: Mannuronan grafted with l-Tyrosine methyl ester and thereafter epimerized.

**Figure 3 gels-05-00023-f003:**
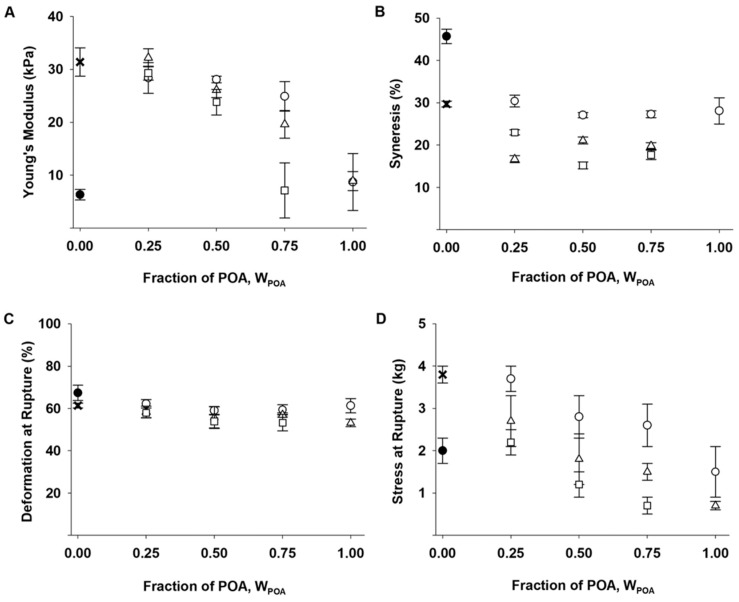
Mechanical characterization of 1.0% *w*/*v* mixed Ca-saturated hydrogels of stipe alginate and POA (*n* = 5–10). (**A**): Young’s modulus (E), (**B**): Syneresis (%), (**C**): Deformation at rupture (%) and (**D**): Stress at rupture (kg) as a function of fraction of POA (W_POA_) for ○: POA P_0_ = 0.02, ∆: POA P_0_ = 0.04 and □: POA P_0_ = 0.08. Gels made of solely POA (W_POA_ = 1.00) is given for P_0_ = 0.02 and P_0_ = 0.04 (not syneresis for POA P_0_ = 0.04). Data for P_0_ = 0.08 is given in Table 2. x: 1.0% (*w*/*v*) stipe alginate gels, • 0.5% (*w*/*v*) stipe alginate gels. Abbreviations: POA = partially oxidized alginate. P_0_ = degree of oxidation.

**Figure 4 gels-05-00023-f004:**
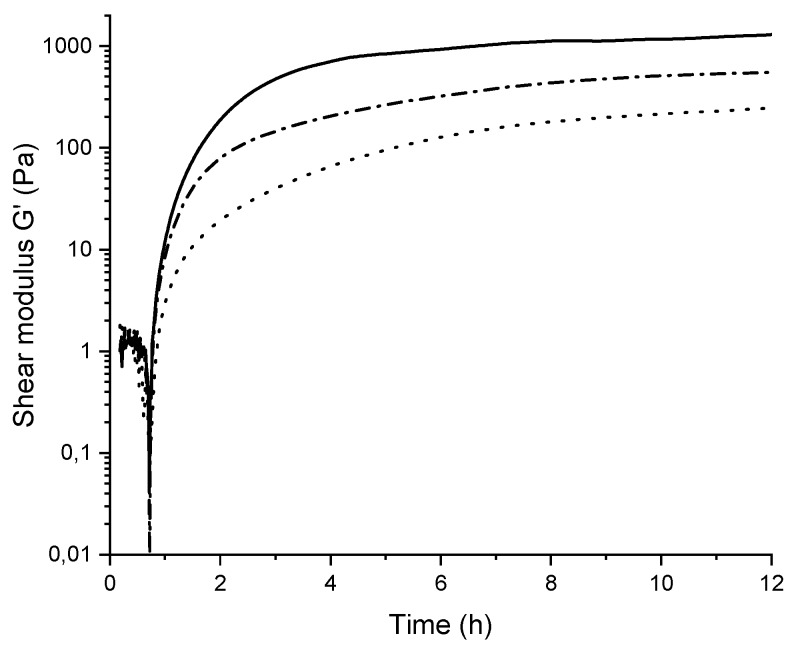
Shear modulus over time for 1.0% *w*/*v* mixed stipe alginate and functionalized POA (P_0_ = 0.08) upon mixture with CaCO_3_ and GDL (at 0 h, 15 mM CaCO_3_), resulting in unsaturated Ca–alginate gels. Fraction of POA, W_POA_ = 0.00 (solid line), fraction of POA, WP_OA_ = 0.50 (-.-.-.-.-.-.-), fraction of POA, W_POA_ = 0.75 (……). POA = partially oxidized alginate.

**Figure 5 gels-05-00023-f005:**
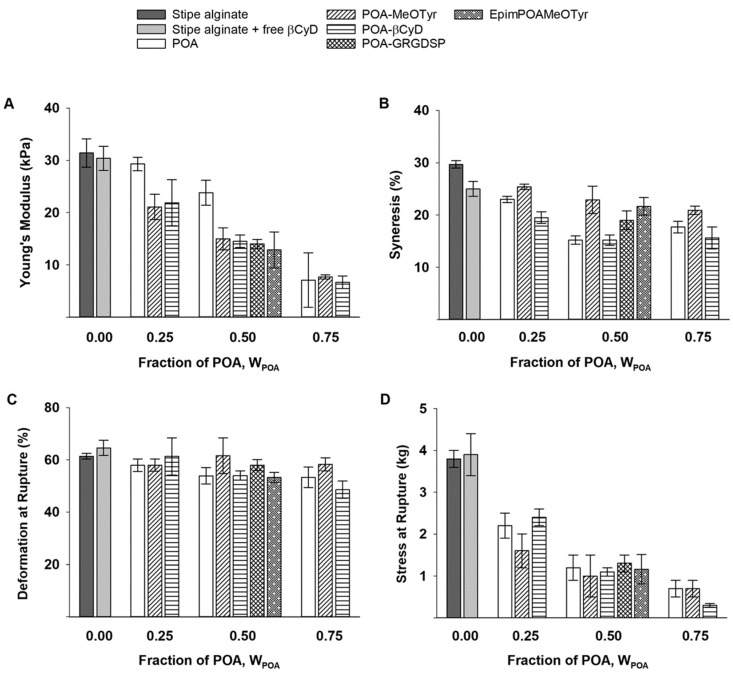
Mechanical characterization of 1.0% *w*/*v* mixed Ca-saturated hydrogels of stipe alginate and functionalized POA (P_0_ = 0.08, *n* = 7–10). (**A**): Young’s modulus (E), (**B**): Syneresis (%), (**C**): Deformation at rupture (%) and (**D**): Stress at rupture (kg) at different weight fractions of functionalized POA (W_POA*_). Abbreviations: POA = partially oxidized alginate. βCyD = β-cyclodextrin. POA-MeOTyr = POA grafted with l-Tyrosine methyl ester. POA-βCyD = POA grafted with β-cyclodextrin. POA-GRGDSP = POA grafted with peptide GRGDSP. EpimPOA-MeOTyr = epimerized mannuronan grafted with MeOTyr.

**Figure 6 gels-05-00023-f006:**
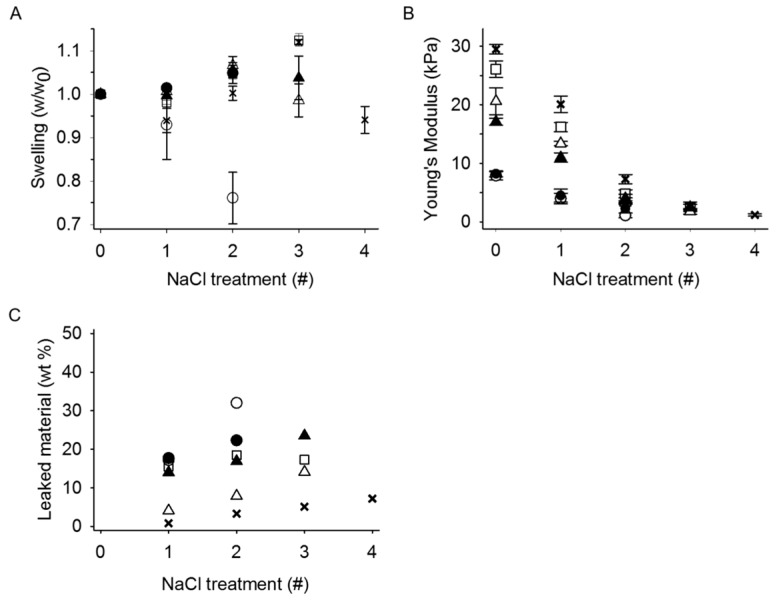
Stability of 1.0% *w*/*v* alginate/POA (P_0_ = 0.08) and alginate/POA-MeOTyr (P_0_ = 0.08, DS = 7.0%) mixed hydrogels (*n* = 5) after consecutive NaCl treatments. One NaCl treatment: immersing the gels in 0.15 M NaCl for 24 h. (**A**): Young’s modulus (E), (**B**): Swelling (*w*/*w_0_*) and (**C**): leaked material (wt%) relative to theoretical total mass of alginate in the gels. Data points at zero saline treatment represents the initial values before immersing the gels in the first saline bath. x: *L. hyperborea* stipe alginate, □: alginate/POA W_POA_ = 0.25, ∆: alginate/POA W_POA_ = 0.50 ○: alginate/POA W_POA_ = 0.75, ▲: alginate/POA-MeOTyr W_POA–MeOTyr_ = 0.50 ●: alginate/POA-MeOTyr W_POA–MeOTyr_ = 0.75. Abbreviations: POA = partially oxidized alginate. POA-MeOTyr = POA grafted with l-Tyrosine methyl ester.

**Figure 7 gels-05-00023-f007:**
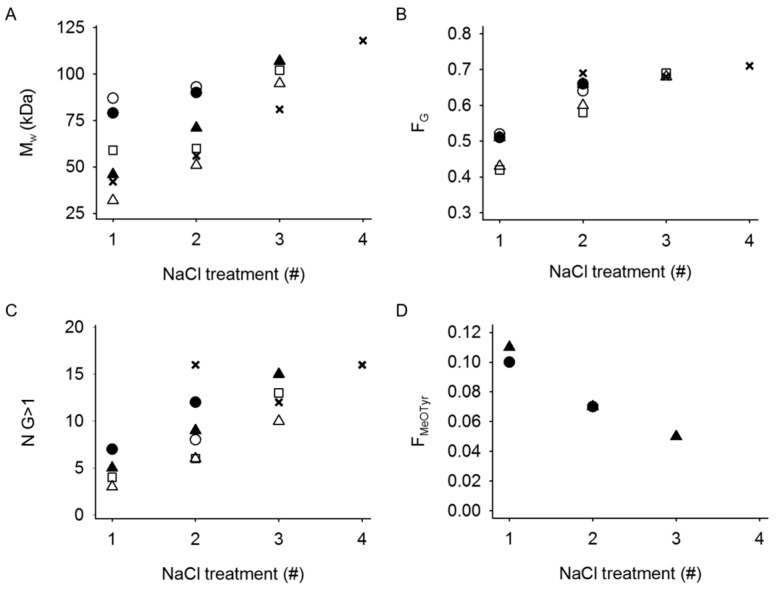
Characterization of leaked alginate from 1.0% *w*/*v* alginate/POA (*P*_0_ = 0.08) and alginate/POA-MeOTyr (P_0_ = 0.08) mixed hydrogels (*n* = 5) after immersing the gels in consecutive solutions of 0.15 M NaCl. (**A**): Weight average molecular weight (*M_w_*), (**B**): Fraction of G residues (F_G_), (**C**): Average G-block length larger than one (N_G>1_) and (**D**): Fraction of MeOTyr (F_MeoTyr_) x: *L. hyperborea* stipe alginate, □: alginate/POA W_POA_ = 0.25, ∆: alginate/POA W_POA_ = 0.50 ○: alginate/POA W_POA_ = 0.75, ▲: alginate/POA-MeOTyr W_POA-MeOTyr_ = 0.50, ●: alginate/POA-MeOTyr W_POA-MeOTyr_ = 0.75. Abbreviations: POA = partially oxidized alginate. POA-MeOTyr = POA grafted with l-Tyrosine methyl ester.

**Table 1 gels-05-00023-t001:** Characterization of the functionalized alginates in this study: P_0_ = mole periodate per mole uronate residue), degree of substitution (DS, mole substituent per mole uronate residue%), average molecular weight (*M*_w_). POA: Oxidized alginate. POA-MeOTyr: Oxidized alginate grafted with methyl tyrosine ester. POA-β-CyD: Oxidized alginate grafted with beta cyclodextrin, POA-GRGDSP: Oxidized alginate grafted with the hexapeptide GRGDSP. Epim POA-MeOTyr: oxidized mannuronan grafted with methyl tyrosine ester and, thereafter, epimerized to a G-content of 49%.

Material	P_0_	DS (%)	M_w_ (kDa)
*L. hyperborea* stipe alginate	0.00	**-**	133
POA	0.02	-	99
POA	0.04	-	93
POA	0.08	-	97
POA-MeOTyr	0.08	7.0	114
POA-β-CyD	0.08	1.6	65
POA-GRGDSP	0.08	3.9	134
Epim POA-MeOTyr	0.08	7.9	126

**Table 2 gels-05-00023-t002:** Mechanical properties of Ca-saturated alginate hydrogels (1.0% *w*/*v*) of modified material solely, based on 8% oxidized stipe alginate (POA), grafted with MeOTyr (POA-MeOTyr), β-CyD (POA-β-CyD) or epimerized alginate grafted with MeOTyr (EpimPOA-MeOTyr).

Alginate	POA	POA-MeOTyr	POA-β-CyD	Epim POA-MeOTyr
Young’s modulus, E (kPa)	1.2 ± 0.5	1.9 ± 0.7	0.7 ± 0.1	2.5 ± 0.3
Syneresis (%)	67 ± 7	46 ± 6	59 ± 3	54 ± 1
Stress at rupture (kg)	0.29 ± 0.06	0.55 ± 0.04	0.13 ± 0.01	3.01 ± 0.04
Deformation at rupture (%)	52 ± 3	60 ± 2	46 ± 3	63 ± 2

**Table 3 gels-05-00023-t003:** Chemical characterization of the starting material used for chemical modification and as unmodified alginate in alginate hydrogels. Composition is described by the fraction of M and G (F_M_ and F_G_), fraction of diad and triplet sequences (e.g., F_GGG_: fraction of alginate consisting of guluronic triplets) and the average length of G-blocks (N_G>1_).

Alginate	M_w_ (kDa)	F_G_	F_M_	F_GG_	F_MG_/F_GM_	F_MM_	F_MGG_/F_GGM_	F_MGM_	F_GGG_	N_G>1_
*L. hyperborea* stipe alginate	133	0.65	0.35	0.53	0.12	0.23	0.05	0.10	0.48	11

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
