# Peer review of "Mechanical Properties of Ca-Saturated Hydrogels with Functionalized Alginate"

_gels, 2019, doi:10.3390/gels5020023_

Round 1
Reviewer 1 Report
In the manuscript titled “Mechanical properties of Ca-saturated hydrogels with
3 functionalized alginate”. An important topic in the field of tissue engineering. The authors characterized hydrogels composed of alginate and modified alginate. The modification entailed using the relatively new periodate oxidation method to attach a cells adhesion peptide and β-cyclodextrin to the alginate chains.
The authors synthesized 7 molecule types in order to examine the effect of the modification on the gel’s mechanical properties. The amount of data presented in this manuscript is impressive. The authors revised the manuscript and I now find it suitable for publication.
Author Response
We thank the reviewer for taking the time to review our paper and for giving comments that has improved our manuscript.
Reviewer 2 Report
This paper describes mechanical properties of calcium-saturated hydrogels with functionalized alginate.
I found that the introduction part of this paper was very informative, authors presented information on hydrogels for biomaterial engineering, tissue regeneration and drug release. However, I did not see clear formulation of the objective of this paper. What is that authors want to achieve? Control over mechanical properties, degradation rate or controlled release of drugs from these hydrogels. Final objective needs more clarification.
In introduction authors claim that they developed new alternative method for functionalization of alginate using partial oxidation with periodate, following reductive amination. However, this technique is no way new, as it was introduced by Gomez et al (2007), where they synthesized oxidized alginates at room temperature during 24 h, in aqueous solution of sodium alginate, using sodium periodate in the dark.
Reductive-amination of oxidized alginate is also widely used to introduce chemical modifications to wide variety of polymers, including alginates (see Andresen, Painter, & Smidsrød, 1977; Carré, Delestre, Hubert, & Dellacherie, 1991; Kang, Jeon, Lee, & Yang, 2002). Please include all these references, if not already.
In Results 2.1 Preparation and Characterization of functionalized hydrogels, authors mention epimerization of grafted mannuronan using AlgE64. I would advise authors to provide more detailed information and possibly scheme of epimization and the reason for this particular modification (increase G content and possibly improve mechanical properties?).
Also Section 2.1 of Results introduces different types of hydrogels with 2-8% degrees of oxidation, following statement that 8% oxidized alginate was selected for further grafting modification. In Section 2.2 authors report on mechanical properties of these hydrogels (table 2), clearly showing that hydrogels with 8% oxidation degree have very poor mechanical properties. Why did authors chose 8% oxidation alginate polymer to proceed with grafting?
Table 2 can possibly have several mistakes (syneresis of POA is given at 67%, while Figure 3 shows value of around 20-45%). I would like to ask authors to add all data points from Table 2 to Figure 3 for reader convenience.
Please clarify what is the difference between Ca-limited and Ca-saturated hydrogels and give more explanation on process of Ca release in presence of GDL.
Authors claim that “Ca-limited gels were chosen due to limited syneresis and hence good contact between the gel and probe”, not providing data to support their statement. Please add data on syneresis of Ca-limited hydrogels.
Please prepare all graphs in this paper with mirrored axes and annotations of all symbols on the graph, not in caption.
I would suggest to extend rheological measurements of shear modulus to be extended over 24 hours, as it is clear from the picture that values of G’ of 50 and 75% POA hydrogels were increasing after 12 hours. Please change in caption WPOA=0 to solid line to correspond the graph data. If possible provide both G’ and G” data, with indication of gelation time for each type of hydrogel.
All bar graphs should have significant difference information, based on t-tests mentioned in the manuscript. This information should also be included in all Figure captions.
Section 2.5 discusses swelling of modified alginate hydrogels, which is increase of volume of polymer network, not “weight gain”. Please correct the swelling definition in manuscript.
Based on stability data, 4 days of physiological salt conditions will completely degrade alginate based hydrogels with any type of oxidation, grafting, mixing ratio. Based on these results, do authors consider these materials feasible for bioengineering applications? Are there any strategies proposed to increase the degradation time of alginate hydrogels?
Author Response
This paper describes mechanical properties of calcium-saturated hydrogels with functionalized alginate.
I found that the introduction part of this paper was very informative, authors presented information on hydrogels for biomaterial engineering, tissue regeneration and drug release. However, I did not see clear formulation of the objective of this paper. What is that authors want to achieve? Control over mechanical properties, degradation rate or controlled release of drugs from these hydrogels. Final objective needs more clarification.
The objective of the paper is now clarified changing the sentence in the introduction from “This study investigates… “ to “The objective of this study is to further investigate… the mechanical properties and stability of hydrogels containing oxidized and functionalized alginates alone and in combination with a strong gel-forming unmodified alginate.”
In introduction authors claim that they developed new alternative method for functionalization of alginate using partial oxidation with periodate, following reductive amination. However, this technique is no way new, as it was introduced by Gomez et al (2007), where they synthesized oxidized alginates at room temperature during 24 h, in aqueous solution of sodium alginate, using sodium periodate in the dark.
This reference is already included, please see reference 30.
Reductive-amination of oxidized alginate is also widely used to introduce chemical modifications to wide variety of polymers, including alginates (see Andresen, Painter, & Smidsrød, 1977; Carré, Delestre, Hubert, & Dellacherie, 1991; Kang, Jeon, Lee, & Yang, 2002). Please include all these references, if not already.
These references have now been included. Please see references 22-24. These authors have used both partial oxidation and reductive amination on alginate, but have used the toxic reducing agents sodium cyanoborohydride (NaBH3CN) or sodium borohydride (NaBH4) (whereas we have used 2-methylpyridine borane complex). We have included this information in the last paragraph of the introduction.
In Results 2.1 Preparation and Characterization of functionalized hydrogels, authors mention epimerization of grafted mannuronan using AlgE64. I would advise authors to provide more detailed information and possibly scheme of epimization and the reason for this particular modification (increase G content and possibly improve mechanical properties?).
The reason for the use of this chemoenzymatic strategy was the hypothesis that this would give better mechanical properties compared to the non-epimerized POA-MeOTyr, due to the grafting only taking place on the M-residues. We have now stated this more clearly in our manuscript, please see the middle of the last paragraph in section 3.1. mechanical properties. A detailed study of the use of AlgE64 on the epimerization of mannuronan, as well as a general scheme of the epimerase, can be found in reference 34 (Stanisci, A.; Aarstad, O. A.; Tøndervik, A.; Sletta, H.; Dypås, L. B.; Skjåk-Bræk, G.; Aachmann, F. L. Overall size of mannuronan C5-Epimerases influences their ability to epimerize modified alginates and alginate gels. Carbohydr. Polym. 2018, 180, 256–263, doi:10.1016/J.CARBPOL.2017.09.094). A general scheme of epimerization of chemically modified mannuronan can be found in reference 20 (Sandvig, I.; Karstensen, K.; Rokstad, A. M.; Aachmann, F. L.; Formo, K.; Sandvig, A.; Skjåk-Bræk, G.; Strand, B. L. RGD-peptide modified alginate by a chemoenzymatic strategy for tissue engineering applications. J. Biomed. Mater. Res. Part A 2015, 103, 896–906, doi:https://doi.org/10.1002/jbm.a.35230). Hence, our opinion is that a scheme of epimerization is already presented in the literature and would prefer not to include it in this paper.
Also Section 2.1 of Results introduces different types of hydrogels with 2-8% degrees of oxidation, following statement that 8% oxidized alginate was selected for further grafting modification. In Section 2.2 authors report on mechanical properties of these hydrogels (table 2), clearly showing that hydrogels with 8% oxidation degree have very poor mechanical properties. Why did authors chose 8% oxidation alginate polymer to proceed with grafting?
The purpose of this study was not to necessarily use only modified alginate to make hydrogels, but to see if a mix of modified alginate and non-modified alginate could be used to create a gel with functionality as well as give tunable mechanical properties. We chose to use 8% oxidation for making the grafted alginates because we hypothesized that this would show us the largest variations between the mixed gels, and thereby help us understand how the grafted alginates would affect the mechanical properties of mixed hydrogels, and if it would be possible to utilize these types of gels in e.g. tissue engineering where for example a difference in mechanical properties can affect stem cell differentiation. If one in addition can modify some of the alginate in the gels with e.g. cell attaching peptides one would be able to create specific environments for different types of cells. We have expanded the first paragraph in the discussion to make our choice clearer for the reader.
Table 2 can possibly have several mistakes (syneresis of POA is given at 67%, while Figure 3 shows value of around 20-45%). I would like to ask authors to add all data points from Table 2 to Figure 3 for reader convenience.
Table 2 shows data for 8% oxidized alginate (these gels are not mixed with non-modified alginate). Figure 3 shows data for gels made of 2 and 4% POA (these gels are not mixed with non-modified alginate), and 2, 4 and 8% POA gels that are mixed in various ratios with non-modified alginate. We clarify this in the figure caption. We have kept the data for the 8% gels separate because we wanted to highlight the differences between the pure 8% oxidized gels and the other gels (mixed or pure 2/4% gels).
Please clarify what is the difference between Ca-limited and Ca-saturated hydrogels and give more explanation on process of Ca release in presence of GDL.
More explanation on the calcium ion release from CaCO3 upon presence of GDL is now added to the materials and methods section: “Briefly, the release of proton from GDL and the subsequent lowering of pH destabilize the CaCO3and cause release of calcium ions that will bind to the alginate.”
Also, the following text is now added to section 4.3 for clarification of difference between Ca-limited and Ca-saturated gels: “The amount of calcium added is calculated to theoretically crosslink all G present, however, it is shown that the gel strength and syneresis of calcium gels further increase with calcium concentration [12]. Hence, Ca-saturated hydrogels were made by immersing the gel cylinders in 50 mM CaCl2 and 0.2 M NaCl (800 mL for 8 cylinders) for 24 hours at 4 °C, to ensure excess of calcium in the gels.”
Authors claim that “Ca-limited gels were chosen due to limited syneresis and hence good contact between the gel and probe”, not providing data to support their statement. Please add data on syneresis of Ca-limited hydrogels.
Upon syneresis, the alginate gels release water and this water release cause slipping of the probe upon rheological measurements. Normally, this can be observed as lowering of G´ upon increase in calcium concentration, that is not a correct result as seen on compression of gel cylinders. Also a reduction in G´ over time is normally also a sign of syneresis. Hence, rheology measurements are very vulnerable to the slipping effect, and hence, syneresis. Syneresis has not been measured for the alginate gels that were used for rheology measurements since the gels are weak and not easy to handle and this easily result in inaccuracy in syneresis measurements.
Please prepare all graphs in this paper with mirrored axes and annotations of all symbols on the graph, not in caption.
We have not changed this, as it is not specified in the template we have from this journal. We also feel that due to the large amount of data, it would be more difficult to understand the figures if the annotations were included in the graphs instead of in the caption.
I would suggest to extend rheological measurements of shear modulus to be extended over 24 hours, as it is clear from the picture that values of G’ of 50 and 75% POA hydrogels were increasing after 12 hours. Please change in caption WPOA=0 to solid line to correspond the graph data. If possible provide both G’ and G” data, with indication of gelation time for each type of hydrogel.
The rheology measurements have not been extended to 24 hours since the small volume gels used in this experiment easily dries out and this introduces errors over prolonged time of measurements. We have now changed the captions according to the reviewers notice. We have not included the G” datain this paper and further not more focus on the time of gelation. As the gelation time is firstly depending on the release of proton from GDL and subsequent release of calcium ions from CaCO3, differences between the mixed gels are secondary to the effects of the calcium release.
All bar graphs should have significant difference information, based on t-tests mentioned in the manuscript. This information should also be included in all Figure captions.
We have already tried to include the significant difference information in the figures, but found that the readability suffered due to too much information in one figure. We therefore decided to include statistical information in a separate appendix.
Section 2.5 discusses swelling of modified alginate hydrogels, which is increase of volume of polymer network, not “weight gain”. Please correct the swelling definition in manuscript.
We have now clarified the definition of swelling in the materials and methods section to: ”Swelling is defined as volume increase, however, as alginate gels increase in volume by absorbing the surrounding solution that correspond to subsequent increase in weight, swelling of the gels were characterized as w/w0, where w0is the initial weight before the first saline treatment and w is the weight after the respective saline treatments.”
Based on stability data, 4 days of physiological salt conditions will completely degrade alginate based hydrogels with any type of oxidation, grafting, mixing ratio. Based on these results, do authors consider these materials feasible for bioengineering applications? Are there any strategies proposed to increase the degradation time of alginate hydrogels?
We have expanded on section 3.3. stability and leakage, to include this in our discussion. The stability of the alginate hydrogel will be affected by other aspects than the type of alginate used to make the gel, and the optimal stability will depend on the purpose of application. For instance, the gelling ion can be changed to give a more robust gel, while pre-treatment in saline bath before placement of the hydrogel will give faster disintegration of the gel.
Please see all changes in document highlighted in the attached document.

Round 2
Reviewer 2 Report
Authors considered all comments and made significant changes to the manuscript, therefore I recommend publication in the present form.